# Albumin: A Review of Market Trends, Purification Methods, and Biomedical Innovations

**DOI:** 10.3390/cimb47050303

**Published:** 2025-04-26

**Authors:** Muhammad Awais Ashraf, Bei Shen, Muhammad Asif Raza, Zhu Yang, Muhammad Nabeel Amjad, Ghayyas ud Din, Lihuan Yue, Afifa Kousar, Qudsia Kanwal, Yihong Hu

**Affiliations:** 1CAS Key Laboratory of Molecular Virology & Immunology, Institutional Center for Shared Technologies and Facilities, Pathogen Discovery and Preservation Platform, Shanghai Institute of Immunity and Infection, Chinese Academy of Sciences, No. 320, Yueyang Road, Shanghai 200031, China; awais@siii.cas.cn (M.A.A.); bshen@siii.cas.cn (B.S.); raza@siii.cas.cn (M.A.R.); 203612304@st.usst.edu.cn (Z.Y.); nabeel@siii.cas.cn (M.N.A.); ghayyas@siii.cas.cn (G.u.D.); lhyue@siii.cas.cn (L.Y.); 2University of Chinese Academy of Sciences, Beijing 101408, China; 3Institute of Bismuth Science, University of Shanghai for Science and Technology, Shanghai 200093, China; 4Department of Chemistry, The University of Lahore, Lahore 54590, Pakistan; kousarafifa@gmail.com

**Keywords:** albumin, rising demand of albumin, market size, purification techniques, novel applications

## Abstract

Albumin is the most abundant plasma protein, accounting for approximately 50% of total serum protein in healthy individuals. In recent years, albumin has attracted significant attention due to its biocompatibility, non-toxicity (metabolizing in vivo into harmless degradation products), non-immunogenic properties, ease of purification, and water solubility. These characteristics render it an ideal candidate for a wide range of biomedical applications. Its uses include drug delivery systems, wound healing, antioxidant therapies, infusion treatments, COVID-19 therapeutics, tissue engineering, and other critical care domains. Consequently, the global demand for albumin has been steadily increasing. The international albumin market was valued at USD 5394.9 million in 2021 and is projected to reach USD 9192 million by 2030, with a compound annual growth rate (CAGR) of 6.1%. Given its diverse applications and rising demand, substantial efforts have been made to ensure a sustainable supply of albumin. This review provides an overview of albumin, along with its novel applications, purification methods, and market trends.

## 1. Introduction

Proteins are essential macromolecules that perform a wide range of functions, from catalyzing biochemical reactions to providing structural support within cells. Among the extensive family of proteins, albumin stands out due to its unique properties and widespread use in clinical and industrial applications. Human serum albumin (HSA), the most abundant plasma protein, plays a key role in maintaining vascular oncotic pressure (force exerted by plasma proteins, mainly albumin, to retain water in blood vessels and prevent fluid leakage into tissues), regulating lipid metabolism, and serving as a carrier for various molecules. Its biocompatibility, non-immunogenicity, and metabolic safety have made HSA a preferred choice in therapeutic applications, including the treatment of hypovolemic shock, burns, and hypoproteinemia [1,2,3,4,5]. In recent years, albumin has attracted attention for its potential as a drug delivery system. Its capacity to form stable nanoparticles combined with its non-toxic and non-antigenic properties makes albumin-based nanoparticles an appealing platform for therapeutic delivery. These nanoparticles offer advantages such as enhanced stability during in vivo administration, easy scalability, and efficient drug encapsulation [6,7]. Furthermore, albumin’s ability to be metabolized into harmless degradation products enhances its safety profile, making it an ideal candidate for use in injectable drug delivery systems [8,9].

The term ‘albumin’ originates from the Latin word ‘albus’. It is the most abundant plasma protein, known for binding various molecules and regulating vascular oncotic pressure and lipid metabolism [10,11]. HSA is used to treat conditions like hypovolemic shock, burn therapy, and hypoproteinemia. Additionally, HSA also enhances glutathione levels in peripheral blood lymphocytes, which are crucial for normal lymphocyte function and immune responses [12,13]. Moreover, its established lack of immunogenicity makes albumin an ideal candidate for use in vaccine development and other therapeutic applications [14]. This review will focus on albumin’s unique biomedical properties, its diverse applications in drug delivery, and its emerging role in innovative medical therapies. Furthermore, we will summarize the current trends in albumin purification methods and its growing market relevance.

## 2. Types of Albumin

Albumin is classified into the following three major types: ovalbumin (OVA), human serum albumin (HSA), and bovine serum albumin (BSA). OVA is a monomeric phosphoglycoprotein, commonly consumed as food, with 385 amino acid residues and a disulfide bond structure. The isoelectric point (pI) of OVA is 4.8, with a molecular weight of 47,000 Da [15]. In comparison, BSA has a molecular weight of 69,324 Da and a pI of 4.7; moreover, it shares some characteristics with HSA and is frequently used in drug delivery applications due to its stability and binding properties [16]. However, HSA is preferred for clinical applications over BSA because it poses a lower risk of inducing immunologic reactions in vivo [17,18]. HSA, being a hydrophilic plasma protein, has a half-life of 19 days, a molecular weight of 64,438 Da, and a pI of 5.9. It contains 35 cysteine residues, including 17 disulfide bridges, which are crucial for its functionality by maintaining protein stability, participating in antioxidant reactions, and facilitating the binding of small molecules and drugs [19]. The features of those three major types of albumin are summarized in Table 1.

These cysteine residues play a crucial role in the formation of three isoforms of HSA, which are classified according to the redox state of cysteine residue at position 34, as follows: mercaptalbumin (HMA), non-mercaptalbumin-1 (HNA1), and non-mercaptalbumin-2 (HNA2) [31]. The redox states of albumin have been linked to various physiological conditions, including aging and nutritional status. Furthermore, research suggests that oxidized albumin is implicated in various diseases, such as liver disease, renal failure, diabetes mellitus, cardiovascular diseases, pulmonary cancer, and Parkinson’s disease [32]. These properties highlight HSA’s potential not only as a drug carrier but also as a biomarker for disease diagnosis and progression.

## 3. Metabolite of Albumin

Albumin is primarily synthesized in the liver, where hepatocytes produce approximately 10–15 g per day in a healthy adult. Its synthesis is regulated by various factors, including nutritional status, colloid osmotic pressure, and hormonal signals, such as insulin and glucocorticoids. Once secreted into the bloodstream, albumin is widely distributed, with about 40% circulating in the plasma and 60% residing in the interstitial space [33]. The degradation of albumin occurs mainly within endothelial cells, hepatocytes, and kidney proximal tubules, with a normal half-life of approximately 19 days in humans [34]. It is taken up through receptor-mediated endocytosis and subsequently degraded in lysosomes. The balance between synthesis and degradation maintains plasma albumin levels under physiological conditions.

The relatively long half-life of albumin enables the body to maintain a stable supply of this essential protein, even amidst fluctuations in its synthesis and degradation. The liver’s capacity to synthesize albumin is crucial for ensuring an adequate and consistent supply of this protein to support various physiological functions [33]. Factors that affect liver function, such as liver disease or nutritional status, can influence albumin synthesis and, consequently, serum albumin levels [35]. The relationship between inflammation, albumin half-life, and the hepatic synthetic rate is complex and varies based on the nature and duration of the inflammatory condition. Acute inflammation typically increases the demand for albumin, leading to a shorter half-life but an increased synthetic rate. Conversely, chronic inflammation may impair albumin synthesis and result in a prolonged half-life. In cases of chronic inflammation, inflammatory states reduce albumin synthesis due to the liver prioritizing acute-phase protein production. However, chronic inflammation also reduces albumin turnover by decreasing its degradation rate, thereby extending its half-life. This phenomenon is believed to be due to altered endothelial and lysosomal processing that slow down the normal clearance of albumin.

Furthermore, the liver’s response to inflammation is a dynamic process aimed at maintaining homeostasis. In the proximal tubules of the human kidney, receptor-mediated endocytosis plays a pivotal role in the selective reabsorption of proteins, including albumin, which are initially filtered into the renal tubules. Specialized receptors on the luminal surface of tubular cells, such as the megalin–cubilin complex, specifically bind to filtered proteins, forming protein–receptor complexes. These complexes are subsequently internalized through endocytosis, transported into the tubular cells within vesicles, and processed in lysosomes. This process ensures efficient reabsorption of valuable proteins and nutrients, preventing their loss in the urine and helping to maintain the body’s overall protein and nutrient balance. Additionally, it helps to prevent conditions such as proteinuria and ensures the body’s efficient use of these vital resources [36,37].

## 4. Functions of Albumin

Albumin is a multifunctional plasma protein that plays a critical role in regulating oncotic pressure. Changes in oncotic pressure can lead to oxidative stress, which may stimulate HSA binding. Due to its unique three-dimensional structure, HSA can bind to a wide range of molecules, including metabolites, gases, and exogenous substances such as drugs. To date, no specific receptor for HSA has been identified; however, it is believed that receptor-mediated endocytosis is the primary mechanism for the uptake of HSA-bound molecules. Additionally, HSA functions as a molecular chaperone, assisting in protein folding and preventing the formation of disease-related aggregated proteins [38,39]. It is also renowned for its antioxidant properties, which help to restore colloid osmotic pressure (COP) and maintain fluid equilibrium between compartments [40]. Hypoalbuminemia treatment typically involves the administration of 5%, 20%, or 25% albumin, which are sterilized at 60.5 °C for 10–11 h to mitigate the risk of viral infection. However, this deactivation process may result in denatured HSA, potentially affecting its functionality [41,42,43,44]. The emergency treatment of acute liver failure, cardiopulmonary bypass, and hypovolemic shock are all clinical indications for albumin 4-5% solutions. During intravenous administration, protein-rich fluids must be carefully managed to ensure proper sequestration [45]. While the clinical application of albumin continues to evolve, it has a wide range of uses in research and analytical fields, including protein stabilization, cryopreservation, and cell culture supplementation. Novel applications of HSA include its use as a drug nanocarrier, oxygen transporter, and peptide fusion agent [9,46]. Nanoparticle delivery systems are extensively researched for their potential in drug delivery. Typically, nanocarriers can protect medications from degradation, enhance drug absorption by facilitating epithelial permeation, alter pharmacokinetic distribution profiles, and promote intracellular uptake and distribution. Moreover, optimal drug delivery systems and biodistribution can be achieved by controlling the surface qualities, organization, and environmental conditions [47,48,49]. One of the significant advantages of nanoparticle frameworks is their capability to withstand physiological stress and enhance organic stability, along with the feasibility of oral delivery. This makes them a superior choice for drug delivery compared to liposomes [50,51]. Various types of nanoparticles are being developed for diverse drug delivery applications, including polymeric nanoparticles, ceramic nanoparticles, nanocages of strong lipid nanoparticles, polymeric micelles, nanowires, polymer–drug forms, nanotubes, and dendrimers, among others, which are being made for various medication conveyance applications [52]. Polymeric nanoparticles can be synthesized from polysaccharides, proteins, or synthetic polymers [53,54,55]. Nanoparticles derived from natural hydrophilic polymers have demonstrated improved biocompatibility, enhanced drug loading capability, and potentially reduced opsonization by the reticuloendothelial system (RES) due to an aqueous steric barrier. Protein-based systems, such as gelatine, collagen, whey protein, albumin, and casein, have been investigated for delivering drugs, nutrients, and bioactive compounds. Probiotic species and peptide proteins serve as excellent raw materials. These materials combine the benefits of synthetic polymers with low degradation toxicity and the high absorbability of final products [56,57,58,59]. The functional summary of albumin is shown in Figure 1.

## 5. Novel Applications of Albumin

Albumin plays a pivotal role in a variety of medical and therapeutic applications, including drug delivery, wound healing, as an antioxidant, infusion therapy, COVID-19 infection treatment, tissue engineering, critical illness treatment, as a drug carrier, respiratory distress syndrome treatment, abdominal and cardiac surgeries, acute brain injury treatment, and cirrhosis treatment. In Figure 2, we classify and illustrate its primary applications.

### 5.1. Drug Delivery

Albumin has been extensively investigated for its potential in drug delivery due to its unique properties, such as biocompatibility, the ability to bind a wide variety of substances, and prolonged circulation time. However, despite the promising outcomes observed in albumin-based drug delivery systems, several challenges need to be addressed to optimize their efficacy. For example, although horse serum albumin has been crystallized to create an effective delivery system for therapeutic agents [60,61], stability and reproducibility issues remain. The structural integrity of albumin can be compromised under certain conditions, which may adversely affect its drug-binding capacity and release profiles. Additionally, albumin-based drug delivery systems, such as Reimazolam-bovine serum albumin nanoparticles (RBNPs) designed for anesthesia, offer controlled release and enhanced stability. Nevertheless, ensuring consistent bioavailability and achieving predictable pharmacokinetics remain active areas of research [62].

Other drug delivery systems, such as poly-(methyl methacrylate, or PMMA) nanoparticles, conjugated with albumin to enhance biocompatibility, present significant potential in cancer therapy [63]. While these systems improve cellular uptake and reduce cytotoxicity, ensuring the long-term stability of albumin-conjugated nanoparticles in vivo and addressing potential immunogenicity concerns remain ongoing challenges. Moreover, regulatory considerations regarding the clinical translation of albumin-based nanomedicines require careful scrutiny. Issues related to batch-to-batch variability, long-term safety, and the approval process for new formulations must be addressed in order to facilitate their widespread clinical applications.

Additionally, albumin-based drug-protecting emulsion systems, such as CoFe_2_O_4_@Albumin nanoparticles, have been developed for hyperthermia treatment [64], demonstrating the potential of harnessing albumin’s versatility in innovative therapeutic applications. Despite these advancements, challenges related to scaling up production, ensuring cost-effectiveness, and maintaining consistency in therapeutic outcomes must be resolved for broader application in clinical settings.

### 5.2. Wound Healing

This section highlights albumin’s critical role in accelerating wound healing. Albumin is essential for synthesizing collagen tissue, which forms the foundation for wound healing and cell regeneration. Additionally, it helps to maintain osmotic pressure, delivers nutrients to cells, and promotes tissue regeneration, including bone growth [65,66]. Albumin actively aids in the healing of various types of wounds, such as ocular injuries and postoperative wounds [67,68,69,70]. Bovine serum albumin (BSA) can be combined with Polycaprolactone (PCL) to create biohybrid nanofibrous wound dressings, enhancing PCL’s quality and mechanical attributes. These PCL-BSA dressings exhibit greater flexibility and accelerate cell proliferation, facilitating faster cell regeneration [71]. Nanofibrous wound dressings constructed from the synthetic polymer polyacrylonitrile (PAN) and BSA represent another novel strategy. A wound dressing composed of albumin, silver, and glycerol has also been developed for infected wounds. Even when subjected to random rolling and rotation, this glycated bovine serum albumin (G-BSA) wound dressing maintains its integrity and exhibits excellent flexibility and strength. It demonstrates a 227% increase in breaking strength, with a tensile strength of 0.46 MPa.

Initially, G-BSA lacks antibacterial properties. This limitation is addressed by converting the G-BSA film into G-BSA-Ag by immersion in a silver solution at 4 °C for 6 h, which significantly inhibits bacterial growth. The biocompatibility and non-toxicity of BSA are crucial for the controlled release of silver ions in the films, preventing disruptions to metabolic pathways and adverse effects [72].

### 5.3. Antioxidant

Furthermore, researchers have investigated the antioxidant properties of albumin, which are attributed to its ligand-binding capacity (ability interact with the molecule). Wu et al. (2020) demonstrated that BSA exhibits antioxidant activity when exposed to monascin (MS), as evidenced by the 2,2-diphenyl-1-picrylhydrazyl (DPPH) assay. This interaction leads to the formation of BSA-MS complexes at site I, effectively reducing oxidative stress levels [73,74]. Additionally, another study highlighted the effective antioxidant activity of albumin in combination with coumestrol, particularly in scavenging hydroxyl radicals [75,76]. Huo et al. (2020) further advanced the understanding of the antioxidant activity of Firenze glycoprotein (FPZ) on BSA fibrillation, as assessed through DPPH and ABTS assays. Notably, the IC50 values for ABTS and DPPH were 0.092 mg/mL and 0.249 mg/mL, respectively, indicating potent antioxidant properties. These findings suggest that FPZ, with its active binding sites, can effectively function as an antioxidant. Such antioxidant activity may have significant implications for delaying cellular aging and enhancing cell viability [77]. In summary, albumin’s multifaceted properties encompass its role as a powerful antioxidant.

### 5.4. Infusion Therapy

For over five decades, albumin infusion has been a standard procedure, playing a vital role in the treatment of various medical and surgical problems [78]. HSA infusion has been particularly beneficial in clinical situations such as acute liver failure, burns, shock, surgery, cardiopulmonary bypass, acute respiratory distress syndrome, and trauma [79]. Notably, Tansil Tan and Nathasia applied albumin infusion therapy to pediatric and adult patients with toxic epidermal necrolysis (TEN), continuing it for 20 days. They observed promising outcomes in avoiding TEN progression and reducing patient mortality [80]. Additionally, Violi et al. (2021) demonstrated that albumin can accelerate skin regeneration in a compelling study [81]. This study investigated the effects of albumin infusion in COVID-19 hypercoagulation patients. Among 29 patients with pneumonia, with elevated D-dimer plasma levels (>1 mg/L) and lower albumin serum levels (<3.5 g/dL), 10 patients received seven days of albumin infusions, while the remaining 19 patients served as the control group. Both groups were administered low-dose heparin. Notably, the mean age of the albumin group was 82 years, compared to 73 years in the control group. The findings suggest that albumin may possess anticoagulant properties that influence plasma D-dimer levels. However, concerns remain regarding the use of human serum albumin (HSA) infusions, including the risk of viral contamination, limited donor availability, and high production costs, which increase patient expenses. Nurdiansyah et al. proposed exploring alternative sources of albumin beyond HSA, such as mammals, reptiles, amphibians, apes, and fish, to address these problems [79].

### 5.5. Therapeutic Strategy for COVID-19 Infection

Liu et al. conducted a focused study in Shenzhen, China, involving twelve patients, eight of whom were male, with the majority being over 60 years old. The objective of the study was to identify clinical markers that could predict the severity and progression of COVID-19. The researchers found that several laboratory parameters were indicative of a poor prognosis. Specifically, they observed that patients with hypoalbuminemia, characterized by lower levels of albumin, had a higher risk of severe disease outcomes. Additionally, these patients exhibited lymphopenia, defined as a reduction in lymphocyte count, alongside elevated levels of C-reactive protein (CRP) and lactate dehydrogenase (LDH), which are markers of inflammation and tissue damage [82].

In their study, the researchers also quantified angiotensin II concentrations using enzyme-linked immunosorbent assay (ELISA) techniques. They found that angiotensin II levels were significantly elevated in these patients, and this increase was correlated with higher viral loads and more extensive lung injury. The findings suggested that an imbalance in the renin–angiotensin system (RAS) may contribute to the severe respiratory complications seen in COVID-19 patients. This led the researchers to hypothesize that treatment strategies targeting the RAS, such as the use of ACE inhibitors or angiotensin receptor blockers (ARBs), might offer therapeutic benefits in managing COVID-19 [82].

Ongoing research supports the potential benefits of serum albumin therapy for COVID-19 patients. This is based on the complex interplay between albumin levels and the expression of ACE2 receptors, which are crucial for viral entry into host cells [83]. If albumin could be utilized to deliver bioactive compounds such as epigallocatechin gallate (EGCG) and curcumin—which possess antiviral properties—directly to the intracellular environment, it could potentially enhance treatment efficacy. Moreover, when combined with medications designed to block viral fusion or entry into host cells, this strategy might represent a promising approach for treating SARS-CoV-2 infections [84,85].

The spike glycoproteins on the surface of the SARS-CoV-2 virus, which confer its characteristic crown-like appearance, play a crucial role in the viral entry into host cells. These spike proteins contain receptor-binding domains (RBDs) that bind with high affinity to the ACE2 receptors on the surface of human cells. This interaction is essential for the virus’s ability to enter the cell and initiate infection, significantly enhancing the pathogenicity of SARS-CoV-2 [86]. Once the spike protein binds to the ACE2 receptor, the virus is internalized by the host cell through endocytosis. This binding and subsequent internalization leads to a downregulation of ACE2 receptors on the cell surface, potentially exacerbating disease severity [87].

ACE2, or angiotensin-converting enzyme 2, is a membrane-bound enzyme that plays a vital role in the regulation of the renin–angiotensin system. It is predominantly expressed in the epithelial cells of the lungs, heart, kidneys, intestines, and blood vessels, where it serves as a key modulator of cardiovascular and respiratory function [88]. By converting angiotensin II into angiotensin 1–7, ACE2 acts as a counter-regulatory mechanism within the RAS, mitigating the effects of angiotensin II, which include vasoconstriction, inflammation, and oxidative stress [89]. Angiotensin 1–7 exerts its effects by binding to the Mas receptor, a G-protein-coupled receptor that mediates vasodilation, reduces thrombosis, and possesses anti-inflammatory and antiproliferative properties [90]. This protective role of ACE2 and its downstream signaling pathway underscores the potential therapeutic implications of modulating the RAS in the context of COVID-19.

### 5.6. Tissue Engineering

In the field of tissue engineering, albumin has emerged as a versatile biomaterial, recognized for its biocompatibility, ability to modulate immune responses, and potential to support cellular function. For example, Hsu et al. [91] demonstrated the application of an albumin-based fibrous scaffold in neural tissue engineering. The incorporation of hemin into the scaffold enhanced the attachment and differentiation of neural stem cells, thereby supporting tissue repair and axonal regeneration. This demonstrates the potential of albumin-based materials to not only provide structural support, but also to offer functional assistance in tissue regeneration. However, while albumin-based scaffolds are beneficial for specific applications, their performance may vary depending on the tissue type and the complexity of the tissue engineering requirements. Comparatively, other biomaterials, such as collagen, gelatin, and synthetic polymers, have also been extensively studied within this domain. Collagen is often regarded as a gold standard for soft tissue regeneration due to its natural abundance in the extracellular matrix. However, collagen-based materials can suffer from poor mechanical properties and limited stability, especially when subjected to enzymatic degradation. In contrast, albumin-based scaffolds demonstrate enhanced stability and reduced degradation rates, making them a more durable alternative in certain contexts. Nevertheless, although albumin’s mechanical strength is adequate for soft tissues, it may prove insufficient for the development of more rigid tissues such as bone. Furthermore, concerns regarding supply limitations, as it is primarily derived from human or bovine serum, raise concerns about supply and ethical considerations, which may limit its scalability.

Albumin coatings on implant surfaces have been shown to reduce bacterial colonization, as demonstrated by Cometta et al. [92], who found that immobilizing HSA onto scaffold surfaces significantly reduced Staphylococcus aureus colonization. This anti-biofilm activity is an important feature for tissue engineering applications, as biofilm-related infections are a major challenge in implantable medical devices. However, compared to other antimicrobial coatings, such as those made from silver or antibiotic-loaded materials, albumin may exhibit less potent antibacterial effects. While albumin’s ability to modulate the immune response is beneficial for reducing inflammatory reactions and enhancing biocompatibility, its immunomodulatory effects might not be sufficient to address all inflammatory challenges in tissue engineering, particularly in the context of chronic inflammation. The immunocompatibility of albumin is further highlighted by the work of Tao et al. [93], who utilized albumin’s surface-passivation effect to reduce immune rejection in xenografts. This application underscores the significance of albumin in improving the biocompatibility of foreign tissues, potentially leading to safer and more effective tissue engineering solutions. However, although albumin aids in reducing immune rejection, it may not fully overcome the immunological challenges presented by xenogeneic tissues. In this regard, synthetic biomaterials such as poly-(lactic-co-glycolic acid) (PLGA) and polyethylene glycol (PEG) hydrogels have been employed to improve tissue integration and reduce immune responses. These synthetic materials offer more controlled properties and greater design versatility, allowing for more tailored approaches in tissue engineering.

Thus, while albumin demonstrates significant potential in tissue engineering applications, it is essential to carefully evaluate it alongside other biomaterials. This evaluation should ensure the most suitable selection for a specific application, taking into account factors such as the material’s mechanical properties, degradation rates, immunocompatibility, and scalability.

### 5.7. Critical Illness

Albumin infusion has been considered a potential treatment option for sepsis, septic shock, and fluid resuscitation in critically ill patients admitted to the intensive care unit (ICU) for several decades. The use of albumin is based on its properties as a colloid, which can expand plasma volume more effectively than crystalloids, potentially improving hemodynamic stability in critically ill patients. However, the clinical benefits of albumin infusion, particularly in comparison to other fluid replacements, such as crystalloids, have been the subject of extensive research, leading to varying and sometimes conflicting conclusions.

One of the most significant studies in this area is the Saline versus Albumin Fluid Evaluation (SAFE) trial, conducted approximately two decades ago. The SAFE trial was a large, multicenter, randomized controlled trial that enrolled critically ill patients admitted to the ICU. The trial compared the outcomes of patients who received 4% albumin solution with those who received normal saline. The primary findings indicated no significant difference between the two groups in terms of 28-day mortality, organ failure, or ICU length of stay [94]. However, a post hoc analysis focusing specifically on patients with severe sepsis revealed that albumin therapy was associated with a reduced odds ratio of death [95], suggesting a potential benefit in this subgroup of patients.

Building on the findings from the SAFE trial, the Albumin Italian Outcome Sepsis (ALBIOS) trial further investigated the role of albumin in patients with sepsis or septic shock. In this trial, patients were treated with a 20% albumin solution, titrated to achieve a serum albumin concentration of 30 g/dL, in combination with crystalloids, versus crystalloids alone. The results showed no significant difference in 90-day survival between the two groups. However, similar to the SAFE trial, a post hoc analysis of patients with septic shock indicated that albumin treatment was associated with lower 90-day mortality [96], highlighting a potential survival benefit in this specific patient population.

Further research has also examined the use of albumin in other critically ill populations. For instance, a randomized controlled trial (RCT) conducted in septic cancer patients admitted to the ICU evaluated the effects of administering a bolus of albumin in lactated Ringer’s solution versus lactated Ringer’s solution alone during the initial six hours of fluid resuscitation. The study found no significant improvement in mortality at either 7 or 28 days [97], suggesting that albumin may not confer a mortality benefit in this subgroup of critically ill patients.

Several meta-analyses have been conducted to synthesize the findings from these and other studies on the use of albumin-containing solutions for resuscitation in critically ill patients. Over time, these meta-analyses have yielded conflicting results, with some studies suggesting potential benefits while others have found no significant advantages compared to the use of crystalloids [98,99]. Recent meta-analyses have generally concluded that, although albumin administration is safe, it does not provide a clear survival benefit over crystalloids. These mixed results have fueled ongoing debates about the role of albumin in fluid resuscitation for sepsis and septic shock.

Given the existing evidence and considering the high cost of albumin relative to crystalloids, as well as the safety concerns associated with the use of starches, the 2021 International Guidelines of the Surviving Sepsis Campaign do not recommend albumin infusion as the first choice for fluid resuscitation in patients with sepsis or septic shock. Instead, the guidelines suggest—with a low level of evidence—that albumin may be considered in patients who have received large volumes of crystalloids [100]. Additionally, albumin may be preferred in patients who cannot tolerate high volumes of crystalloids, such as those with cirrhosis. In these patients, albumin solutions (5% or 20%) are thought to be more effective than crystalloids in reversing arterial hypotension during septic shock, although the impact on short-term survival remains inconsistent across studies [101].

### 5.8. Drug Carrier

Albumin has emerged as a promising drug carrier, offering numerous biological applications due to its unique properties. As a naturally occurring plasma protein, albumin is biocompatible, non-immunogenic, and biodegradable, making it an excellent candidate for drug delivery systems [102]. Its long half-life of approximately 19 days allows drugs bound to or encapsulated within it to circulate for extended periods, thereby enhancing therapeutic efficacy and reducing the frequency of dosing [103]. One of its key advantages is its ability to accumulate in tumor tissues via the enhanced permeability and retention (EPR) effect, making albumin particularly suitable for targeted cancer therapies [104]. Additionally, albumin’s versatility in binding both hydrophilic and hydrophobic drugs improves their solubility and stability, leading to controlled drug release and reduced systemic toxicity [102]. However, albumin-based systems are not without limitations. Premature dissociation of drugs from albumin in circulation may result in suboptimal delivery, while variability in drug binding capacity can lead to inconsistent loading and release profiles. Furthermore, despite benefiting from the EPR effect, albumin lacks inherent tissue specificity, which may necessitate additional modifications, such as ligand conjugation, to enhance targeting [105]. Albumin serves as a drug carrier through multiple mechanisms that enhance drug stability, solubility, and targeted delivery. The high-affinity binding sites on albumin allow for non-covalent drug binding, particularly at Sudlow’s site I and II, which enable the transport of hydrophobic drugs such as paclitaxel, doxorubicin, and methotrexate [106]. In addition to passive drug binding, albumin can be chemically conjugated with therapeutic agents to improve their half-life and bioavailability. One of the key advantages of albumin in drug delivery is its ability to undergo receptor-mediated transcytosis. Albumin interacts with specific cell surface receptors, such as gp60 (albondin), leading to endothelial transcytosis and enhancing drug penetration across biological barriers [107]. Furthermore, albumin is recognized by the neonatal Fc receptor (FcRn), which protects it from lysosomal degradation, thereby extending its circulatory half-life and improving drug retention [108].

In targeted drug delivery applications, albumin takes advantage of the enhanced permeability and retention (EPR) effect observed in tumor tissues, allowing for passive accumulation of albumin-bound drugs at cancer sites [109]. Additionally, albumin nanoparticles can be surface-functionalized with targeting ligands such as antibodies, peptides, and folic acid to improve tumor specificity and receptor-mediated uptake. Advanced formulations such as albumin-coated nanoparticles, albumin–micelle hybrids, and albumin fusion proteins are currently being developed to further optimize drug delivery efficiency.

Real-world applications of albumin-based drug carriers include Abraxane (albumin-bound paclitaxel), which significantly enhances the solubility and bioavailability of paclitaxel, thereby reducing the reliance on toxic solvents like Cremophor EL [110]. Other promising developments include albumin–insulin conjugates for diabetes management, which leverage albumin’s prolonged half-life to achieve sustained glucose control, as well as albumin-bound photosensitizers for photodynamic cancer therapy.

When compared to other drug delivery systems such as polymeric nanoparticles, liposomes, dendrimers, and inorganic nanoparticles, albumin stands out for its superior biocompatibility and safety profile, particularly in long-term applications. However, synthetic carriers like polymeric nanoparticles and dendrimers may offer greater flexibility in surface modification and more precise targeting [111]. While liposomes and inorganic nanoparticles are effective for drug encapsulation and stability, they can encounter challenges such as rapid clearance and potential immune activation, issues that are less prevalent with albumin. The success of albumin-based systems is evident in real-world applications like Abraxane, an albumin-bound paclitaxel formulation that has shown improved efficacy and reduced toxicity in cancer therapy compared to its conventional counterpart [112]. Additionally, albumin–drug conjugates have demonstrated promising outcomes in the treatment of chronic disease, such as albumin–insulin conjugates for diabetes management, by providing prolonged glucose control [113].

### 5.9. Respiratory Distress Syndrome

Acute respiratory distress syndrome (ARDS) is a critical lung condition characterized by fluid accumulation in the lungs, impairing oxygen transport throughout the body and making breathing difficult. The increased permeability of the alveolar–capillary membrane allows protein-rich fluid to infiltrate the alveoli. This disruption, combined with surfactant inactivation and alveolar exudates, results in severe hypoxemia, decreased lung compliance, and inefficient carbon dioxide elimination. Although comprehensive research is limited, the existing literature suggests that administering HSA to ARDS patients with hypoproteinemia can enhance lung function [114,115,116,117]. Colloidal solutions are favored over crystalloids for reducing alveolar–capillary permeability, limiting inflammatory cell infiltration, and minimizing tissue damage. However, synthetic colloids like gelatin and hydroxyethyl starch may cause renal injury and sepsis, increasing mortality risk. Consequently, clinical studies recommend HSA for ARDS management due to its favorable safety and efficacy profile.

### 5.10. Abdominal and Cardiac Surgeries

Patients undergoing abdominal surgeries often experience hypermetabolic conditions, leading to negative nitrogen balance and reduced anabolic activity. The body’s inflammatory response to surgical trauma exacerbates capillary leakage and vascular permeability, lowering serum albumin (Ab) concentrations and frequently resulting in hypoproteinemia. A European study involving 138 patients highlighted a threefold increase in postoperative complications when serum Ab levels decreased by 10 g/L or more on the first postoperative day. Monitoring these levels helps to identify high-risk patients for postoperative complications [118]. Additionally, low Ab levels are common in abdominal surgeries involving pancreatic fistulas. A study of 247 pancreaticoduodenectomy patients identified the postoperative Ab concentration as a significant risk factor for pancreatic fistula development. Managing Ab levels preoperatively may reduce infection risks, as shown in research involving 268 patients, where preoperative Ab was an independent predictor of intra-abdominal infections following hepatectomy. Clinical trials recommend maintaining Ab levels above 30 g/L in critically ill abdominal surgery patients to minimize complications [119,120].

During cardiac surgery, fluid management is crucial due to hypoproteinemia and hemodynamic instability caused by cardiopulmonary bypass [121]. Factors such as blood dilution, ischemia–reperfusion injury, and the release of inflammatory mediators further complicate recovery. Studies indicate that HSA effectively stabilizes fluid balance and reduces the need for transfusions compared to hydroxyethyl starch or Ringer’s lactate solution [122]. Research involving 970 patients demonstrated that hydroxyethyl starch led to increased transfusion requirements and blood loss [123]. Combining HSA with crystalloid solutions reduced readmission and mortality rates in cardiopulmonary bypass patients [124]. Therefore, HSA is recommended for managing hypoproteinemia in cardiac surgery, mitigating complications such as renal insufficiency [125,126].

### 5.11. Acute Brain Injury

Acute brain injury (ABI) can result from conditions such as infections, strokes, metabolic encephalopathy, or traumatic brain injury, often leading to disability or death [127]. Although HSA is generally not recommended as a primary resuscitation fluid for ABI patients or for reducing intracranial pressure, its use in cases of cerebral hemorrhage has been associated with improved neurological outcomes [128,129,130]. Clinical guidelines suggest the administration of a 25% HSA infusion to support neurological prognosis in these cases [130,131,132].

### 5.12. Cirrhosis

Decompensated cirrhosis often leads to complications such as hepatic encephalopathy, variceal bleeding, and hepatorenal syndrome (HRS), characterized by severe renal vasoconstriction due to disrupted systemic and splanchnic circulation. Effective management, including perioperative liver transplantation care, significantly impacts survival outcomes [133,134]. Hypoproteinemia is common in cirrhosis, increasing mortality risks [135]. Large-volume paracentesis combined with HSA infusion has been shown to reduce post-procedure complications and improve survival rates. Some studies indicate a 57% reduction in refractory ascites and a 61% reduction in mortality with HSA treatment, though meta-analyses reveal varied survival outcomes based on differences in the control group [136,137]. In HRS patients, combining HSA with terlipressin demonstrates higher recovery rates compared to other treatments [138]. Additional Ab infusions have been shown to reduce the incidence of acute kidney injury and related mortality. Studies have shown that maintaining Ab levels above 30 g/L does not offer significant clinical advantages over standard care [139]. However, careful Ab management remains crucial for improving patient outcomes.

## 6. Albumin Purification

Albumin, a crucial protein from both biological and medical perspectives, undergoes purification using techniques such as chromatography, ultrafiltration, and precipitation. These methods separate albumin from complex mixtures, ensuring high purity and activity levels. By exploiting differences in size, charge, and affinity, these techniques enable precise isolation for research or therapeutic applications. The choice of method depends on factors such as source material and intended use, highlighting the versatility of albumin purification in biomedical science (Figure 3).

### 6.1. Heat-Shock Method

Albumin can be purified using the heat-shock technique due to its high thermal stability compared to other plasma proteins. This method, also known as thermal precipitation or denaturation, leverages the differing thermal stability profiles of proteins. By heating a protein mixture containing albumin to a temperature above the denaturation point of most proteins (typically between 60 and 80 °C), proteins that are less stable will denature and precipitate, while albumin remains soluble. The process involves heating the mixture, incubating it at the desired temperature, rapidly cooling it to stop the denaturation, and then centrifuging it to separate the precipitated proteins from the soluble albumin. The supernatant containing albumin is collected, as albumin can withstand temperatures up to 60 °C, which can also help to inactivate potential pathogens [26]. Caprylic acid (0.04 M) was used to stabilize the pH at 5 during incubation at 60 °C to isolate albumin from the serum. Other serum proteins denature under these condition and precipitate, allowing the liquid albumin to be concentrated using ultrafiltration to achieve a purity of about 98% [140]. The albumin prepared using the heat-shock method exhibits exceptional purity, exceeding 99%, with a yield of 21 g per liter of plasma [141].

### 6.2. Precipitation Using Ammonium Sulfate Salt

An alternative method for albumin purification combines ammonium sulfate precipitation with liquid chromatography, resulting in a purification yield of exceeding 90% for albumin [26]. Albumin is purified via ammonium sulfate precipitation by adjusting the concentration to 50% saturation and incubating for 24 h at 4 °C, causing albumin to precipitate while other proteins remain soluble. The precipitate is then dialyzed against PBS (pH 7.2) and triphosphate buffer (pH 8.2), followed by further purification and analysis using gel electrophoresis and chromatography. Optimal conditions, including ammonium sulfate concentration, pH, and incubation time, are critical for high-purity albumin isolation [142].

### 6.3. Ion Exchange Chromatography (IEC)

Moore and Stein’s groundbreaking work in the 1940s, which earned them a Nobel Prize [143] for separating amino acids on derivatized polystyrene beads cross-linked with divinylbenzene, laid the foundation for ion exchange, one of the oldest and most widely used protein purification techniques [144]. IEC is commonly employed for processing albumin and other proteins [145], with anion exchange being the most prevalent method [146]. Human blood plasma serves as the primary source for obtaining albumin through IEC, achieving an impressive recovery rate of approximately 95% [147]. Ion exchange chromatography (IEC) for albumin purification involves the following three main steps: pre-treatment, purification, and polishing. Pretreatment clarifies the crude mixture through centrifugation or filtration and ensures buffer compatibility. In the purification step, the sample is loaded onto an ion exchange column, where albumin is selectively captured while contaminants are washed away. Polishing further enhances purity through gradient elution, releasing albumin from the resin. The process, illustrated in Figure 4, requires careful optimization of pH, salt concentration, and resin selection. While traditional IEC has limitations like extended manipulation times, a novel ion exchange membrane chromatography (IEM) method, utilizing improved adsorbers, addresses these drawbacks. Frerick et al. compared IEM and IEC through simulation experiments [148]. Membrane chromatography offers several advantages over column-based separations, including higher separation efficiencies due to shorter diffusion periods, reduced buffer consumption, lower floor space requirements, and a streamlined procedure without complex apparatus or packing needs [149,150,151,152]. Despite these benefits, current IEM technology faces challenges, including high membrane costs and limited efficiency for industrial applications [153,154]. Additionally, compared to modern resins, membranes exhibit relatively poor binding capacity for antibodies [155]. Recent advancements in ion exchange chromatography have led to the development of ion exchange membrane chromatography (IEM), which addresses several limitations of traditional IEC. One of the primary improvements offered by IEM is its superior separation efficiency, which can be attributed to reduced diffusion periods and accelerated mass transfer, thereby facilitating shorter processing durations. Moreover, IEM systems require less buffer consumption, making them more cost-effective for large-scale operations. They also occupy a smaller physical footprint, minimizing the necessity for large chromatography columns and offering a more compact solution. The simplified process also eliminates the need for complex equipment, such as packing columns, making it a more streamlined alternative. However, despite these advantages, IEM technology does face challenges, including high membrane costs and limited binding capacities for large proteins, such as antibodies. While IEM offers several advantages, particularly in terms of efficiency and scalability, its broader adoption is somewhat constrained by these cost-related and capacity limitations. Nonetheless, IEM represents a significant step forward in protein purification, especially for applications requiring high throughput and reduced operational costs.

### 6.4. Albumin Purification Using Gel Filtration Chromatography

In gel filtration chromatography for albumin purification, a column packed with porous beads is used to separate molecules based on size. The column is equilibrated with a buffer that maintains albumin stability. The sample is prepared by removing particulates and, if necessary, concentrating it through ultrafiltration or dialysis. Once loaded onto the column, larger molecules such as albumin pass through more quickly, while smaller molecules are slowed down by entering the pores of the beads. Eluted fractions are collected and monitored, typically by measuring absorbance at 280 nm, and albumin fractions are identified using methods such as SDS-PAGE, Bradford assay, or UV spectroscopy. The protein obtained through gel filtration chromatography exhibits exceptional purity, achieving a 99% purity level [156]. Electrophoresis techniques are employed to verify the quality of purification [157]. Pure albumin fractions are pooled, while fractions containing contaminants or unwanted proteins are discarded. If desired, the purified albumin can be concentrated using methods such as ultrafiltration or dialysis. The buffer can be exchanged with the desired storage buffer or an appropriate buffer for downstream applications, if necessary.

### 6.5. Affinity Chromatography

Cuatecasas et al. first developed affinity chromatography, a type of liquid chromatography for purification, in 1968 [158]. The stationary phase consists of a biological material known as an affinity ligand. Identifying or developing an appropriate ligand that specifically binds to albumin is crucial to its purification. Common ligands used for albumin purification include fatty acids, dyes (e.g., trypan blue or bromocresol green), or immobilized metal ions (e.g., copper or zinc) [26]. In affinity chromatography for albumin purification, a ligand specific to albumin is immobilized onto a solid support, such as agarose or Sepharose beads. The column is equilibrated with a buffer that ensures albumin stability. The albumin sample is prepared by removing particulates and adjusting the composition to match the binding conditions. The sample is then loaded onto the column, where albumin binds to the ligand while other proteins flow through. The column is washed to remove unbound proteins, and the bound albumin is eluted using an elution buffer that disrupts the ligand–albumin interaction, tailored to achieve the desired purity and yield. This method is particularly effective for purifying albumin and examining its biological interactions [159,160,161,162]. The recovery rate of albumin by affinity chromatography is approximately 96% [163].

### 6.6. Electrophoresis Analysis

Under reduced conditions, the combination of sodium dodecyl sulfate–polyacrylamide gel electrophoresis (SDS-PAGE) and cellulose acetate membrane electrophoresis (CAME) was employed to access the purity of various chromatography fractions. CAME was performed using a cellulose acetate membrane (70 × 130 mm), and samples were loaded into the cathode area. Electrophoresis was conducted for 20 min at a continuous current of 0.7 mA/cm in a barbital buffer (pH 8.6, 0.05 mol/L barbital sodium). SDS-PAGE was carried out at a constant voltage on a 12% gel. The protein bands were visualized using Coomassie blue staining (Figure 3) [142]. SDS-PAGE and CAME analyses confirmed a high purity level, around 99%, for the purified human albumin [157]. The properties of different purification methods for albumin are summarized in Table 2.

## 7. Albumin Market

As illustrated in Figure 5, the global albumin market was worth USD 5394.9 million in 2021 and is expected to hit USD 9192 million by 2030 with a compound annual growth rate (CAGR) of 6.1% [170]. Albumin, being the most concentrated protein in blood plasma, performs various critical functions, including the transport of fatty acids and hormones, as well as maintaining oncotic pressure. Therapeutically, it is used to restore blood volume and replace lost fluids following wounds, burns, surgeries, liver diseases, and infections. Additionally, albumin helps to stabilize drugs by preventing oxidation, surface absorption, and aggregation. Recombinant albumin is produced using biotechnological methods, primarily through yeast and bacteria strains.

Remarkably, the growth of the global albumin market is influenced by the substantial number of surgical procedures conducted worldwide. According to the World Health Organization (WHO), approximately 235 million major surgeries were performed in 2019. Additionally, data from the American Cancer Society revealed that, in 2020, there were 1,806,590 new cancer cases in the United States, further underscoring the importance of albumin. The increasing number of cancer-related surgeries often leads to hypoalbuminemia, thereby driving market expansion [171]. Albumin market growth is also propelled by increased research and development (R&D) efforts and the expanding use of bovine, human, and recombinant serum albumin. Advancements in protein purification, molecular separation, and utilization of albumin as a nanocarrier in drug delivery—owing to its biocompatibility and non-toxic and biodegradable nature—are key growth factors contributing to market growth [172].

The global albumin market is closely related to product type, application, and geography.

The four primary forms of albumin available are OVA, BSA, HSA, and recombinant HSA (rHSA). Among these, HSA has generated the highest revenue and is projected to exhibit the fastest growth, surpassing other sub-segments (OVA, BSA, and rHSA) during the forecast period. Specifically, HSA, BSA, and rHSA are expected to generate revenues of USD 7545 million, USD 336 million, and USD 285 million, respectively. In 2016, the HSA segment held the largest market share, and this trend is anticipated to continue in the coming years. Human serum albumin shared the highest global market of albumin in 2021, followed by ovalbumin, as shown in Figure 6 [170]. The regulatory policies play a critical role in shaping the market for albumin. The approval process for recombinant albumin products, as well as the increasing demand for safer and more effective therapeutics, is significantly shaped by international regulatory standards. For example, the regulatory requirements set forth by the FDA and EMA regarding recombinant proteins can have a substantial impact on market entry and expansion. Additionally, regulatory hurdles related to biosimilar albumin products also influence market growth. These policies ensure that only high-quality products are permitted in the marketplace, which may lead to increased manufacturing costs but simultaneously promote long-term stability within the market.

Ovalbumin powder, sustainably produced from *Trichoderma reesei*, is positioned to replace chicken egg white protein powder in the food industry [173] and ranks second in market significance. The global market for ovalbumin powder is projected to reach USD 2303 million by 2030, driven by its high-quality protein content and versatility. It finds extensive application in food, dietary supplements, pharmaceuticals, and cosmetics. The growing demand for nutrition-rich products and increasing health awareness are key drivers of its market expansion, making it a crucial ingredient across various sectors. Egg ingredient manufacturers’ research and development efforts are creating new opportunities for this essential protein source [174,175,176].

Simultaneously, the market share of rHSA is projected to increase at a significant CAGR of 11.4%. This growth can be attributed to the expanding applications of recombinant albumin in drug and vaccine formulations, as well as increasing public awareness. The market is segmented by application into drug formulation, therapeutics and vaccination, components of growth media, and other applications. In 2016, the therapeutics segment held the largest revenue share due to an increase in surgical procedures, burn injuries, and trauma cases. Additionally, the 2nd largest segment for medication formulation and vaccines is anticipated to exhibit at the fastest rate during the forecast period, as illustrated in Figure 7 [170].

Furthermore, the global albumin market is segmented into the following regions: Caribbean, Asia–Pacific, North America, Latin America, and Europe. In 2016, North America held the largest market share, followed by Europe. Conversely, the Asia–Pacific region is anticipated to dominate the market during the forecast period due to its substantial population, increasing disposable income, and growing patient awareness of albumin products [177,178].

## 8. Challenges and Solutions

Given its non-toxic, non-immunogenic, and highly biocompatible nature, albumin is increasingly being utilized as a novel drug carrier. Albumin has demonstrated certain distinct benefits, particularly for tumor treatment. However, several challenges remain in the development of albumin-based formulations.

The current methods of albumin production predominantly rely on the use of organic chemicals, which unfortunately leads to the denaturation of albumin during processing. Denaturation alters the natural structure of albumin, rendering it inactive and causing the loss of its essential biological functions. This poses a significant challenge for researchers who require functional albumin, especially for biomedical applications such as drug delivery systems. To address the issue, researchers are actively exploring alternative synthesis methods that minimize or eliminate the need for harsh organic chemicals. One promising avenue is the adoption of “green chemistry” principles, which emphasize the use of non-toxic, environmentally friendly, and biocompatible reagents [179]. For example, ionic liquids and supercritical fluids have emerged as potential candidates for producing albumin-based materials. These substances can serve as gentle solvents that reduce the risk of denaturation, thus preserving the protein’s native structure and function. In addition to green chemistry, other innovative techniques are being investigated. One such method is coacervation, a process that involves the separation of a colloidal solution into two distinct phases, one rich in albumin and the other depleted of it, without the need for harmful chemicals. Coacervation can be fine-tuned to maintain the integrity of albumin, ensuring that it retains its functional properties [180]. These alternative approaches not only aim to produce albumin that remains biologically active, but also align with the growing demand for sustainable and eco-friendly manufacturing processes. By reducing reliance on harmful chemicals, these methods hold the potential to revolutionize albumin production and expand its applicability in various fields, particularly in the development of nanoparticles and other advanced biomedical materials.

In recent years, researchers have increasingly focused on albumin, recognizing its potential in various therapeutic applications. Albumin, a versatile protein with essential functions in the human body, has become a subject of intense study due to its unique properties, such as its ability to bind and transport various substances, maintain oncotic pressure, and exhibit antioxidant effects. This growing interest has led to the publication of numerous studies exploring different aspects of albumin, including its structural characteristics, pharmacokinetics, and potential use as a drug carrier or therapeutic agent in conditions like liver cirrhosis, cancer, and cardiovascular diseases [181]. Despite the abundance of research, only a small fraction of these studies has been successfully translated into clinical practice. The challenge of translating albumin research from the laboratory to the bedside remains significant, highlighting the complexity of converting scientific discoveries into viable therapeutic options. One of the primary obstacles is the gap between preclinical findings and clinical application, often referred to as the “valley of death” in drug development. This gap can be attributed to several factors, including the difficulty in replicating laboratory results in human trials, the stringent regulatory requirements for new therapies, and the high costs associated with bringing a new drug to market [182]. To address these challenges, closer collaboration between researchers and clinicians is essential. By working together, these professionals can ensure that research is conducted with a clear focus on clinical relevance, ultimately leading to more successful outcomes. Emphasizing translational research efforts that bridge the gap between laboratory discoveries and real-world patient care can significantly enhance the likelihood of developing effective albumin-based therapies. This approach involves not only identifying promising research avenues but also designing studies that consider the practicalities of clinical application. Moreover, establishing interdisciplinary teams that include experts in molecular biology, pharmacology, clinical medicine, and regulatory affairs can facilitate the process of translating research into practice. These teams can identify potential hurdles early in the development process, enabling more targeted and efficient approaches to overcoming them. Collaborations between academic institutions and pharmaceutical companies are also crucial, as they can provide the necessary resources and expertise to advance albumin research through the different stages of drug development. In addition to collaboration and translational research, extensive preclinical testing is vital for the successful clinical translation of albumin-based therapies. Utilizing relevant animal models that closely mimic human disease conditions can provide critical insights into the safety and efficacy of new treatments. Rigorous evaluations that consider both the therapeutic benefits and potential risks of albumin-based interventions are necessary to ensure that these therapies meet the high standards required for clinical use.

Ultimately, the successful translation of albumin research into clinical applications will depend on a multifaceted approach that combines scientific innovation, collaborative efforts, and a strong commitment to patient-centered care. By addressing the challenges and leveraging the opportunities presented by albumin research, the scientific community can pave the way for new and effective treatments that improve patient outcomes and advance the field of medicine. To ensure its stability within the complex physiological environment of the body, albumin must be modified and adequately stabilized, due to its structural characteristics. The body contains various proteins and enzymes that can potentially degrade or destabilize albumin, making it imperative to enhance its resilience. Researchers have been investigating multiple modification strategies to improve albumin’s stability under these challenging conditions. One effective strategy is chemical modification, particularly PEGylation. This process involves attaching polyethylene glycol (PEG) chains to the albumin molecule [183]. PEGylation significantly enhances the stability of albumin by increasing its size and shielding it from enzymatic degradation. This modification also prolongs albumin’s circulation time in the bloodstream, making it more effective for therapeutic applications.

Another innovative approach is the use of nanotechnology. By encapsulating albumin within protective coatings, researchers can create a barrier against enzymes and other destabilizing factors in the body. These nanocoatings can be engineered to release albumin in a controlled manner, ensuring that it remains stable and functional for extended periods [184]. Furthermore, researchers can enhance albumin’s stability by fine-tuning its formulation to more closely mimic the physiological conditions of the human body. By optimizing parameters such as pH, ionic strength, and temperature, albumin formulations can be made more compatible with the body’s natural environment, thereby enhancing their effectiveness and stability [185]. This approach may involve adjusting the albumin’s molecular structure or incorporating stabilizing agents that reinforce its resilience against the body’s complex biochemical milieu.

## 9. Conclusions

Albumin, a versatile and widely distributed protein, has established itself as a critical component in the biomedical field due to its remarkable binding capacity, regenerative effects, and compatibility with various therapeutic applications. Despite its abundance in nature and relatively low production cost, the clinical use of albumin faces challenges, including limited natural stocks and the complexity of purifying plant-derived serum albumin. Advanced purification techniques such as the heat-shock method, ammonium sulfate precipitation, gel filtration chromatography, ion exchange chromatography, and affinity chromatography are essential for maintaining its effectiveness and meeting the stringent purity standards required for clinical applications. However, these novel extraction and purification methods remain experimental, with current purity levels and production costs posing significant barriers to widespread clinical adoption.

Albumin’s potential in drug delivery, wound healing, antioxidant therapy, and tissue engineering continues to expand, particularly in the context of critical illnesses such as COVID-19, where the albumin globulin ratio has been identified as a predictor of disease severity and mortality. To fully realize its potential in the biomedical field, further technological advancements and rigorous research are essential. As ongoing studies refine our understanding and application of albumin, it is poised to become an even more integral component of future therapeutic strategies.

## Figures and Tables

**Figure 1 cimb-47-00303-f001:**
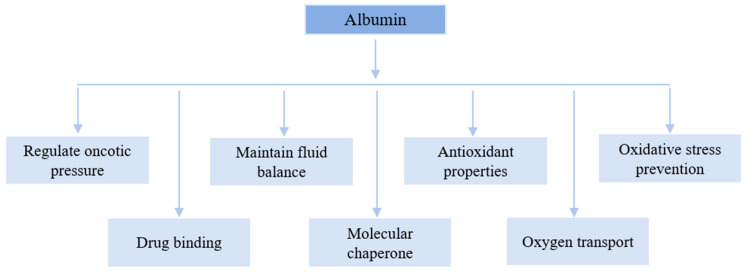
Key functions of albumin. **Note:** Antioxidant properties refer to the intrinsic ability of albumin to neutralize reactive oxygen species (ROS) and reactive nitrogen species (RNS) through its free thiol group at cysteine-34, as well as its capacity to bind transition metals, such as copper and iron, which catalyze the production of free radicals. On the other hand, oxidative stress prevention is a broader concept that encompasses not only the direct scavenging of ROS/RNS by albumin, but also its role in maintaining redox homeostasis. This includes its function in transporting and regulating levels of essential antioxidants (e.g., bilirubin, thiols, and polyunsaturated fatty acids), preventing lipid peroxidation, and modulating inflammatory responses that contribute to oxidative damage.

**Figure 2 cimb-47-00303-f002:**
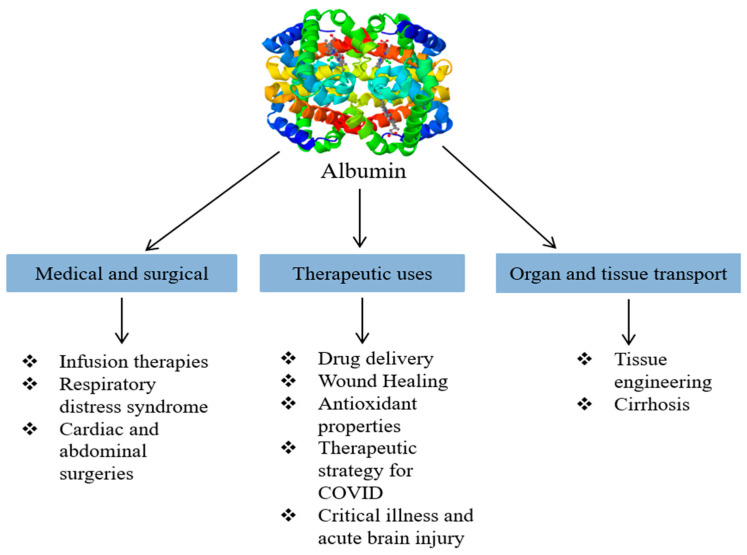
Major applications of albumin.

**Figure 3 cimb-47-00303-f003:**
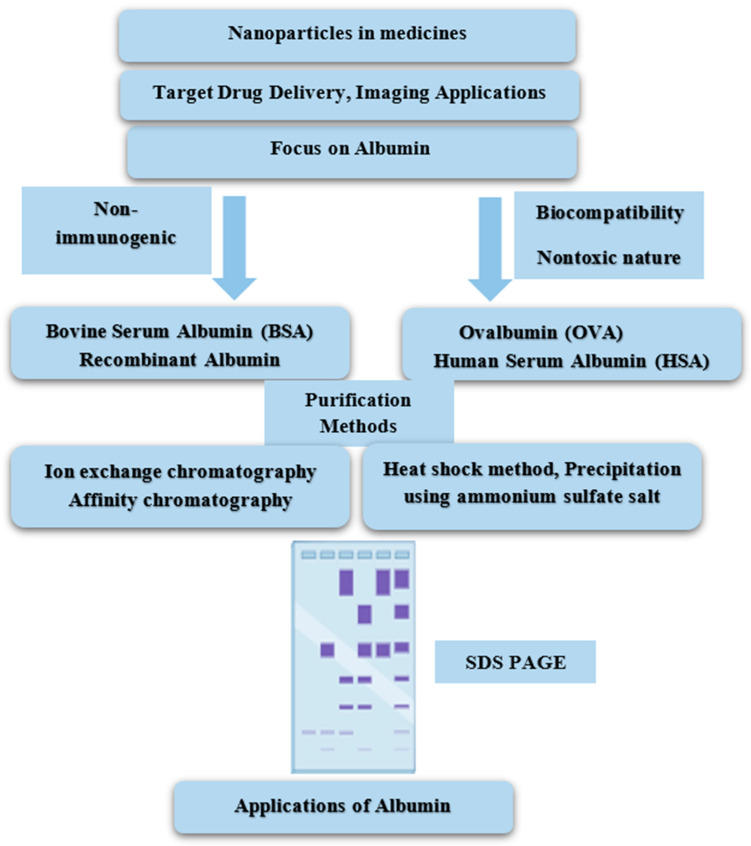
Schematic diagram for isolation and purification of albumin.

**Figure 4 cimb-47-00303-f004:**
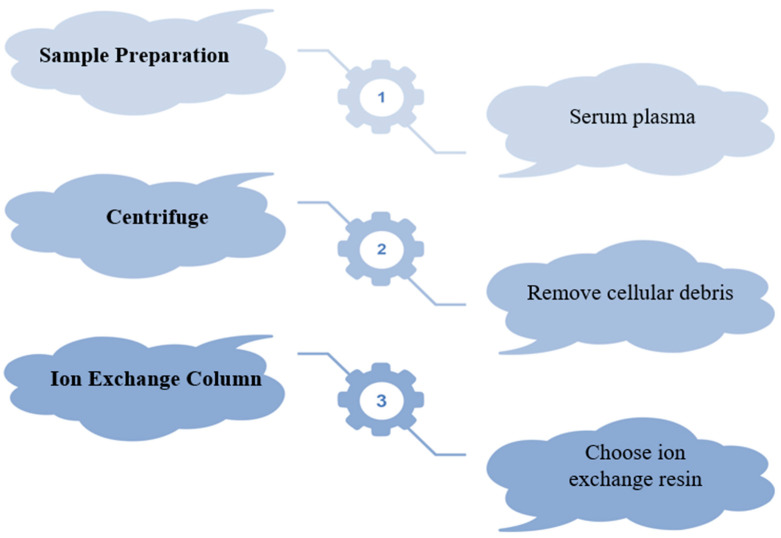
Albumin purification.

**Figure 5 cimb-47-00303-f005:**
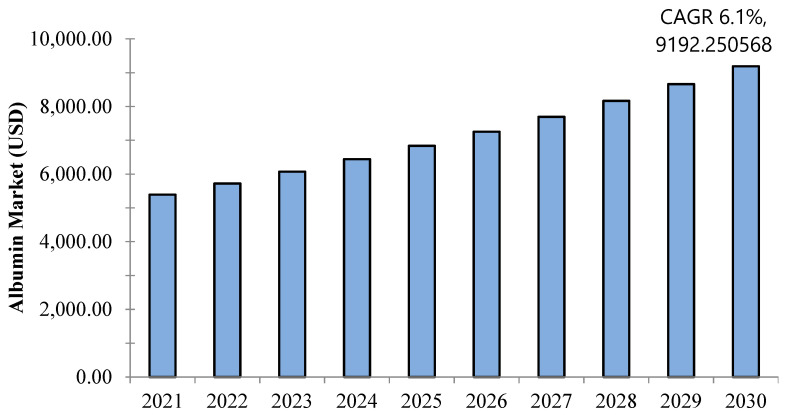
Albumin market forecast for 2021–2030 (CAGR of 6.1% by 2030) [170].

**Figure 6 cimb-47-00303-f006:**
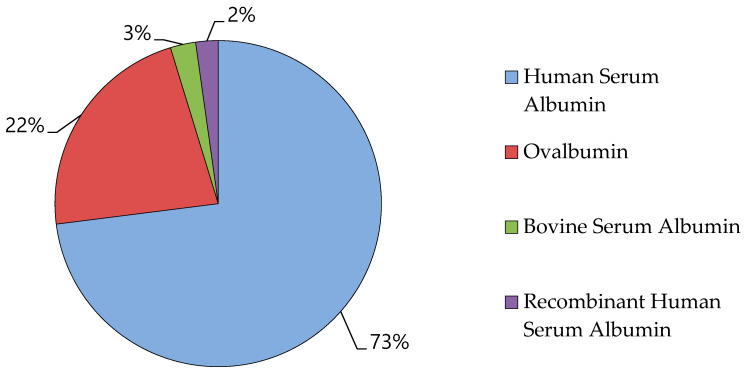
Global market share of various albumin types in 2021.

**Figure 7 cimb-47-00303-f007:**
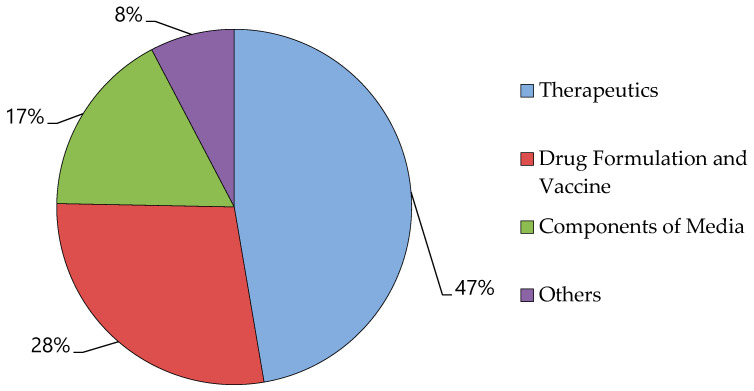
Albumin market by application: therapeutics, drug formulation and vaccine preparation, media manufacturing, and others, respectively [170].

**Table 1 cimb-47-00303-t001:** The major three types of albumin.

Properties	Ovalbumin	Human Serum Albumin	Bovine Serum Albumin	References
Source	Chicken egg white	Human blood plasma	Cow blood plasma	[20,21,22]
Structure	Globular protein, single-chain polypeptide	Globular protein, single-chain polypeptide, heart-shaped structure	Globular protein, single-chain polypeptide, extensively used in labs	[23,24,25]
Molecular Weight	~47 kDa	~64 kDa	~69 kDa	[26]
Isoelectric Point	~4.8	~5.9	~4.7	[26]
Function	Storage protein in eggs as an immunological tool	Maintains osmotic pressure and transports hormones	Carrier protein for hormones and fatty acids	[25,27,28]
Applications	Immunology research, vaccine development	Blood volume expander, drug delivery carrier	Research assays, cell culture diagnostics	[29]
Disadvantages	Potential allergen in some individuals, limited stability in certain conditions	Limited availability and high cost, potential transmission of bloodborne diseases	Potential source of prion transmission, batch-to-batch variability	[30]

**Table 2 cimb-47-00303-t002:** Properties of various albumin purification methods.

Purification Method	Principle	Purity	Application	Disadvantages	Cost	Time Efficiency	Scalability	Reference
Heat-Shock Method	Non-specific denaturation and precipitation of unwanted proteins.	Moderate to high, may have impurities.	Rapid and simple purification for some heat-stable proteins.	Low specificity and limited to heat-stable proteins.	Low	Quick	Limited	[164]
Ammonium Sulfate Precipitation	Solubility differences in proteins at varying salt concentrations.	Moderate to high, depends on conditions.	Initial step for protein concentration, followed by other methods for higher purity.	Non-specific precipitation and challenging to achieve high purity.	Moderate	Moderate	Moderate	[165]
Ion Exchange Chromatography	Differential binding to charged groups on the resin.	High, especially with multiple steps.	Separation of proteins with different charges, effective for highly charged proteins.	Can be harsh for sensitive proteins and pH-sensitive.	High	Moderate	High	[166]
Gel Filtration Chromatography	Separation based on size; smaller molecules take longer to travel through the gel.	High, as it removes smaller impurities.	Effective for desalting, buffer exchange, and separating proteins of different sizes.	Not suitable for very small proteins, might not achieve high resolution.	Moderate	Moderate	Moderate	[167]
Affinity Chromatography	The target protein selectively binds to an immobilized ligand.	High, as it isolates the specific protein.	Excellent for highly specific purification of target proteins.	Ligand selection is critical and can be expensive.	High	Variable	Variable	[168]
Gel Electrophoresis	Movement of charged proteins through a gel matrix based on size and charge.	Depends on the gel and conditions used.	Analyzing protein mixtures, assessing purity, and estimating molecular weight.	Limited scale, time-consuming, and may not provide high purity for preparative purposes.	Low	Variable	Limited	[169]

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
