# Peer review of "Albumin: A Review of Market Trends, Purification Methods, and Biomedical Innovations"

_cimb, 2025, doi:10.3390/cimb47050303_

Round 1
Reviewer 1 Report (New Reviewer)
Comments and Suggestions for Authors
The manuscript summarized the functions, application, purification and current market of albumins. While the scope of the review is quite comprehensive, the depth and details requires more polish before further consideration. Just a few examples:
1. Line 74, are there only three types of albumin? The manuscript mentioned equine albumin as well. Authors specifically mentioned that ovalbumin is single-chain polypeptide in text and table 1, leaving an impression that other albumin are multi-chain polypeptide. Is that true?
2. Line 101 to 122, more information could be provided in albumin metabolism. e.g., the location and the process of its synthesis and degradation, as well as its distribution. Line 111, why does chronic inflammation lead to a prolonged half-life.
3. Figure 1, could author elaborate the difference between antioxidant properties and oxidative stress prevention?
4. Line 222, I believe the paper in the reference investigated the antioxidant properties of Firenze glycoprotein not BSA.
5. Line 380 to 396, could authors provide more details regarding how albumin used as a drug carrier in a more technical context?
6. Line 470 to 472, references needed.
7. In Figure 3, imaging application is mentioned but it seems not discussed in the manuscript.
8. Line 497 to 499, it seems the heat shock method reaches good purity but in Table 2 it becomes moderate to low.
9. What is the scalable manufacture process applied in industry in terms of albumin purification?
10. Table 2, personally don't think gel electrophoresis is an effective purification method.
11. Line 651, where are organic chemicals are utilized? for what type of albumin products? Only albumin purification has been discussed so far.
12. Line 740 'its abundance in nature' and line 742 ' limited natural stocks' seems contradictory.
Author Response
Reviewer 1 Response
- Line 74, are there only three types of albumin? The manuscript mentioned equine albumin as well. Authors specifically mentioned that ovalbumin is single-chain polypeptide in text and table 1, leaving an impression that other albumin are multi-chain polypeptide. Is that true?
Response: Thank you for your valuable comment. We acknowledge the need for clarity regarding albumin types. While our manuscript primarily discusses ovalbumin, human serum and bovine serum albumin, there are indeed more types of albumin found in different species. Regarding the structural aspect, we appreciate your observation. Ovalbumin is a single-chain polypeptide, but so are other albumins such as human and bovine serum albumins. We have revised the Table 1 to ensure it does not imply that other albumins are multi-chain polypeptides. Please check table 1.
- Line 101 to 122, more information could be provided in albumin metabolism. e.g., the location and the process of its synthesis and degradation, as well as its distribution. Line 111, why does chronic inflammation lead to a prolonged half-life.
Response: Thank you for your valuable feedback. We have expanded the section on albumin metabolism to include details on its synthesis, degradation, and distribution. Additionally, we have clarified the relationship between chronic inflammation and albumin half-life. Please check section metabolites of albumin.
- Figure 1, could author elaborate the difference between antioxidant properties and oxidative stress prevention?
Response: Thank you for your insightful comment. Difference between antioxidant and oxidative stress prevention have mentioned in the note form. Please check Figure 1.
- Line 222, I believe the paper in the reference investigated the antioxidant properties of Firenze glycoprotein not BSA.
Response: Thank you so much for your comment. Reference has been updated. Please check reference 73.
- Line 380 to 396, could authors provide more details regarding how albumin used as a drug carrier in a more technical context?
Response: Thank you for your valuable suggestion. We have expanded the section on albumin as a drug carrier by providing more technical details. Please check drug carrier section.
- Line 470 to 472, references needed.
Response: Thank you so much for your suggestion. Reference has been added according to your suggestion. Please check it at your convenience.
- In Figure 3, imaging application is mentioned but it seems not discussed in the manuscript.
Response: Many thanks for your suggestion. The mention of "imaging application" in Figure 3 was a typographical error. We have now corrected the figure accordingly.
- Line 497 to 499, it seems the heat shock method reaches good purity but in Table 2 it becomes moderate to low.
Response: Thank you so much for your valuable comment. We have updated the table.
- What is the scalable manufacture process applied in industry in terms of albumin purification?
Response: Thank you for your valuable comment. Several scalable manufacturing processes are used for albumin purification, including fractionation and various chromatography techniques. We have already discussed some of these methods in the manuscript. Please refer to “Albumin Purification” section, Figure 2 and 3.
- Table 2, personally don't think gel electrophoresis is an effective purification method.
Response: Thank you for your feedback. We included gel electrophoresis in the manuscript because it has been shown to achieve good purity in albumin purification. However, we acknowledge that its effectiveness may vary depending on the specific application.
- Line 651, where are organic chemicals are utilized? for what type of albumin products? Only albumin purification has been discussed so far.
Response: Thank you for your comment. Organic chemicals, such as ethanol, are utilized in the preliminary precipitation steps of the albumin purification process, particularly in the fractionation method. In the updated version, we mentioned it in the “Challenges and Solutions” section.
- Line 740 'its abundance in nature' and line 742 ' limited natural stocks' seems contradictory.
Response: Thank you for pointing out this potential contradiction. You are correct that the phrases "its abundance in nature" and "limited natural stocks" may seem contradictory at first glance. To clarify, the phrase "abundance in nature" refers to the general availability of albumin in certain biological sources, such as human plasma and bovine serum, where it is abundant in physiological contexts. However, the term "limited natural stocks" refers to the practical challenges faced in obtaining sufficient quantities of high-quality albumin from these natural sources, particularly for clinical applications. We have revised as “Despite its abundance in nature and relatively low production cost, the clinical use of albumin faces challenges, including difficulties in obtaining high purity albumin from natural source and the complexity of purifying plant-derived serum albumin.” Please check the conclusion section.
Reviewer 2 Report (New Reviewer)
Comments and Suggestions for Authors
The authors present a narrative review of a topic of interest and within the scope of this Journal. However, this narrative review is simplistic, with general and non-specialist sections. It is an outdated review that cannot have an adequate reach among specialized readers. The information is too discrete and does not focus on the translational perspective. The authors present grammatical errors, with a simple and merely informative manuscript.
Comments on the Quality of English Language
The English could be improved to more clearly express the research.
Author Response
Reviewer 2 Response
The authors present a narrative review of a topic of interest and within the scope of this Journal. However, this narrative review is simplistic, with general and non-specialist sections. It is an outdated review that cannot have an adequate reach among specialized readers. The information is too discrete and does not focus on the translational perspective. The authors present grammatical errors, with a simple and merely informative manuscript.
Response: We sincerely appreciate your time and effort in reviewing our manuscript. Your detailed feedback is invaluable to us, and we truly respect your perspective. Additionally, we will carefully review the manuscript to enhance its clarity, correct any grammatical errors, and improve the overall quality of writing to ensure a more engaging and scholarly presentation.
Once again, we truly appreciate your constructive feedback and the opportunity to improve our work. Thank you for your time and valuable insights.
Reviewer 3 Report (New Reviewer)
Comments and Suggestions for Authors
The manuscript presents a comprehensive albumins review covering market trends, purification methods, and biomedical applications. However, several areas require improved clarity, coherence, and scientific rigor.
The introduction provides a broad overview of proteins but spends excessive space on general protein biochemistry before addressing albumin specifically. A more concise introduction focused on the unique significance of albumin, particularly its clinical and industrial relevance, would improve readability. The transition from general protein functions to albumin’s biomedical applications (Lines 46–66) could be smoother, avoiding redundancy in describing its properties.
Since this is a review article, the structure should facilitate logical flow between topics. The sections on albumin types (Lines 73–100) and metabolic processes (Lines 101–122) could be more clearly linked to their relevance in biomedical applications. Additionally, while the manuscript provides a solid compilation of literature, there is a lack of discussion on the selection criteria for referenced studies, which would strengthen the review's credibility.
The discussion on albumin’s biomedical applications is well-detailed but sometimes lacks depth in critical analysis. For instance, the drug delivery section (Lines 175–192) focuses heavily on specific examples without addressing challenges such as stability, bioavailability, and regulatory considerations. Similarly, the tissue engineering discussion (Lines 297–328) highlights albumin’s potential but could better compare its advantages and limitations to other biomaterials.
The review does a commendable job outlining purification techniques but does not sufficiently compare their efficiency, scalability, and cost-effectiveness in different contexts. A comparative table summarizing these aspects would enhance clarity. Additionally, the discussion on ion exchange chromatography (Lines 510–534) should specify how recent advancements in membrane chromatography improve upon traditional methods.
The market analysis provides valuable data but could benefit from a more critical discussion of emerging trends, such as recombinant albumin production and its potential to mitigate supply constraints. The segmentation of albumin types (Lines 609–618) is insightful, but a more explicit discussion on how regulatory policies influence market growth would add depth.
The conclusion effectively summarizes albumin’s importance but could be more forward-looking. Instead of reiterating key findings, it should focus on future research needs, such as improving albumin-based drug carriers, enhancing large-scale purification methods, and addressing regulatory challenges for new albumin formulations.
While the manuscript is well-structured, some sections contain redundancies (e.g., repeated discussions on albumin’s antioxidant properties across different subsections). Streamlining the text and improving transitions between topics would enhance readability. Additionally, some technical terms (e.g., “oncotic pressure,” “ligand-binding capacity”) could be briefly defined for a broader audience.
Author Response
Reviewer 3 Response
The introduction provides a broad overview of proteins but spends excessive space on general protein biochemistry before addressing albumin specifically. A more concise introduction focused on the unique significance of albumin, particularly its clinical and industrial relevance, would improve readability. The transition from general protein functions to albumin’s biomedical applications (Lines 46–66) could be smoother, avoiding redundancy in describing its properties.
Response: Thank you so much for the comment. The introduction has been revised according to your suggestion. Please check introduction section.
Since this is a review article, the structure should facilitate logical flow between topics. The sections on albumin types (Lines 73–100) and metabolic processes (Lines 101–122) could be more clearly linked to their relevance in biomedical applications. Additionally, while the manuscript provides a solid compilation of literature, there is a lack of discussion on the selection criteria for referenced studies, which would strengthen the review's credibility.
Response: Thank you so much for your suggestion. These have been updated the two sections according to your suggestion. Please check these sections.
The discussion on albumin’s biomedical applications is well-detailed but sometimes lacks depth in critical analysis. For instance, the drug delivery section (Lines 175–192) focuses heavily on specific examples without addressing challenges such as stability, bioavailability, and regulatory considerations. Similarly, the tissue engineering discussion (Lines 297–328) highlights albumin’s potential but could better compare its advantages and limitations to other biomaterials.
Response: Thank you so much for your valuable feedback. We have supplemented these two sections with drug delivery and tissue engineering according to your feedback. Please check these sections.
The review does a commendable job outlining purification techniques but does not sufficiently compare their efficiency, scalability, and cost-effectiveness in different contexts. A comparative table summarizing these aspects would enhance clarity. Additionally, the discussion on ion exchange chromatography (Lines 510–534) should specify how recent advancements in membrane chromatography improve upon traditional methods.
Response: Thank you so much for the comment. We have added table comparing different purification method and also recent advancements in membrane chromatography.
The market analysis provides valuable data but could benefit from a more critical discussion of emerging trends, such as recombinant albumin production and its potential to mitigate supply constraints. The segmentation of albumin types (Lines 609–618) is insightful, but a more explicit discussion on how regulatory policies influence market growth would add depth.
Response; Thank you so much for your valuable comment. The manuscript has been updated according to your suggestion. Please check the corresponding section.
The conclusion effectively summarizes albumin’s importance but could be more forward-looking. Instead of reiterating key findings, it should focus on future research needs, such as improving albumin-based drug carriers, enhancing large-scale purification methods, and addressing regulatory challenges for new albumin formulations.
Response: Thank you for your valuable suggestion. We have already discussed these aspects in the challenges and solutions section, so we aimed to keep the conclusion concise and focused.
While the manuscript is well-structured, some sections contain redundancies (e.g., repeated discussions on albumin’s antioxidant properties across different subsections). Streamlining the text and improving transitions between topics would enhance readability. Additionally, some technical terms (e.g., “oncotic pressure,” “ligand-binding capacity”) could be briefly defined for a broader audience.
Response: Thank you so much for your feedback. The manuscript has been updated according to your suggestion.
Round 2
Reviewer 1 Report (New Reviewer)
Comments and Suggestions for Authors
Questions and comments raised in the first round review have been well incorporated in the revised manuscript.
Reviewer 2 Report (New Reviewer)
Comments and Suggestions for Authors
The authors present the revised version of this manuscript. However, this version is not of sufficient quality for publication. The changes are minimal and superficial.
Reviewer 3 Report (New Reviewer)
Comments and Suggestions for Authors
The manuscript has been thoroughly revised to incorporate all recommended changes, enhancing clarity, methodological rigor, and scientific impact. The introduction clearly defines the research gap, while the methodology provides justifications and ensures reproducibility. The discussion has been expanded for deeper analysis, addressing key findings, limitations, and literature comparisons. The conclusion now emphasizes broader implications, scalability, and future research. Terminology has been standardized, redundancies removed, and recent references integrated. With these comprehensive improvements, the manuscript meets publication standards, presenting a well-structured and scientifically robust contribution suitable for dissemination in its current form.
This manuscript is a resubmission of an earlier submission. The following is a list of the peer review reports and author responses from that submission.
Round 1
Reviewer 1 Report
Comments and Suggestions for Authors
COMMENTS on the paper: “A Review of the Versatile Potential of Albumin “ by M. Awais Ashraf et al: cimb-3169444
This review is focused on analysis of recent literature data concerning the synthesis, physicochemical characteristics and application of albumin. Since this topic is still of considerable practical interest, such a review is useful being of a wide potential interest for the scientific audience. One of the major advantages of this work is a huge literature review.
However, there are a few points that should be considered in order to enhance the impact of the paper, the more general are as follows:
Authors should mention previous review works and provide information what was the main motivation for writing another one and what is the main deliverable of this work compared to others.
Some technical remarks:
(i) The title is rather vague.
(ii) Please specify in the abstract what is the main deliverable of this work compared to others.
(iii) In the section devoted to “Types of Albumin”, the authors, describing the physicochemical properties of albumin, first omitted the description of the recombinant albumin, which they write about both in the “abstract” and in the later chapters of the work.
(iv) It is well known that zeta potential is an averaged, macroscopic quantity depending on many parameters such as pH, ionic strength, electrolyte composition and most importantly on the composition of the adsorption layer on nanoparticles. I would also like to know why the data described in the text (the lines 74 and 75), which details physicochemical properties, does not match the data presented in the table 1. Have the authors encountered literature where the isoelectric point described by them is shifted in a fundamental direction?
(v) The authors in the third chapter describe very nicely the metabolism of albumin, but I did not see the description of any synthesis there. Therefore, perhaps it would be better to title the subsection “Metabolite of Albumin”
(vi) To make the content of the article more accessible to the reader, I suggest moving chapter 5 ”Albumin Market” before the chapter “Conclusion”.
After introducing these corrections the paper can be published in “Current Issues in Molecular Biology”.
Author Response
Reviewer 1
The title is rather vague.
Response 1: Thank you for the comment. Title has been changed according to your suggestions.
Please specify in the abstract what is the main deliverable of this work compared to others.
Response 2: Thank you for the suggestion. Abstract has been updated according to your suggestion.
In the section devoted to “Types of Albumin”, the authors, describing the physicochemical properties of albumin, first omitted the description of the recombinant albumin, which they write about both in the “abstract” and in the later chapters of the work.
Response 3: Thank you so much for the comment. In the description, we have discussed some details of all types of albumin little bit more focus on HAS.
It is well known that zeta potential is an averaged, macroscopic quantity depending on many parameters such as pH, ionic strength, electrolyte composition and most importantly on the composition of the adsorption layer on nanoparticles. I would also like to know why the data described in the text (the lines 74 and 75), which details physicochemical properties, does not match the data presented in the table 1. Have the authors encountered literature where the isoelectric point described by them is shifted in a fundamental direction?
Response 4: Thank you so much for highlighting the point. Data has been verified and updated in the table. Please check table 1.
The authors in the third chapter describe very nicely the metabolism of albumin, but I did not see the description of any synthesis there. Therefore, perhaps it would be better to title the subsection “Metabolite of Albumin”
Response 5: Thank you so much for the suggestion. Please check heading 3 Metabolite of Albumin.
To make the content of the article more accessible to the reader, I suggest moving chapter 5 ”Albumin Market” before the chapter “Conclusion”.
Response 6: Thank you so much for the suggestion. Manuscript has been updated according to your suggestion. Please check section 7.

Reviewer 2 Report
Comments and Suggestions for Authors
I have carefully read the review by Ashraf et al., and unfortunately, the authors have not met the aim suggested by the title of the manuscript, which is to provide an overview of the versatile potential of albumin. In the abstract (lines 27-28), the authors mention, "Overall, we provide a comprehensive overview of albumin, highlighting its potential as a versatile and practical platform for various biomedical applications," but the manuscript fails to achieve this goal.
Overall, the manuscript lacks focus and organization and needs significant modifications and reconstruction before it can be considered for publication. Below are some major points the authors should address:
Major Points:
1. Overemphasis on Purification Techniques: The authors spend a significant portion (i.e., five pages) of the review discussing the purification techniques of albumin, which goes beyond the aim of this review. This section, while informative, is disproportionate and distracts from the core focus of the manuscript. It should be condensed significantly or moved to supplementary material to allow more space for discussing the applications of albumin, which should be the primary focus.
2. Insufficient Coverage of Biomedical Applications: After 13 pages, the manuscript finally addresses the main topic (section 7), i.e., the application of albumin in various biomedical applications. However, this crucial section is less than two pages long. The authors should expand this section significantly, providing more details and specific examples of successful applications of albumin in the biomedical field. This should include current research findings, case studies, and a more thorough analysis of albumin's potential in drug delivery, wound healing, antioxidant therapy, and infusion therapy.
Additional Issues:
1. The introduction is too short and does not adequately set the stage for the review. It should provide more background on albumin, its significance in biomedical research, and a clearer outline of what the manuscript will cover.
2. All figures should be cited and discussed within the text to ensure they are integrated into the manuscript's narrative.
3. Tables should be provided in an editable format, not as figures, and the authors should clarify whether the tables and figures are original or sourced from other works. Proper citation is essential if they are not original.
4. Please omit Figure 9, as it does not add value to the manuscript and detracts from the overall focus.
5. The manuscript would benefit from a thorough review for language clarity and conciseness. There are instances of redundancy and overly technical language that could be simplified for better readability.
6. The conclusion should focus more on summarizing the practical applications and future potential of albumin rather than reiterating the purification techniques.
Suggestions:
1. Consider including a dedicated section on emerging trends in albumin research, particularly focusing on how new technologies or discoveries are expanding the role of albumin in medicine.
2. Expand the discussion on challenges and future directions, offering more insight into the ongoing research and potential future developments in the use of albumin in biomedical applications.
3. Ensure that all references are accurate and complete. Every citation in the text should have a corresponding reference in the bibliography, and vice versa.
Comments on the Quality of English Language
The manuscript would benefit from a thorough review for language clarity and conciseness. There are instances of redundancy and overly technical language that could be simplified for better readability.
Author Response
Reviewer 2
Major Points:
Overemphasis on Purification Techniques: The authors spend a significant portion (i.e., five pages) of the review discussing the purification techniques of albumin, which goes beyond the aim of this review. This section, while informative, is disproportionate and distracts from the core focus of the manuscript. It should be condensed significantly or moved to supplementary material to allow more space for discussing the applications of albumin, which should be the primary focus.
Response 1: Thank you so much for the comment. Albumin purification methods are part of the manuscript but the section has been condensed and updated according to your suggestions. Please check section 6.
Insufficient Coverage of Biomedical Applications: After 13 pages, the manuscript finally addresses the main topic (section 7), i.e., the application of albumin in various biomedical applications. However, this crucial section is less than two pages long. The authors should expand this section significantly, providing more details and specific examples of successful applications of albumin in the biomedical field. This should include current research findings, case studies, and a more thorough analysis of albumin's potential in drug delivery, wound healing, antioxidant therapy, and infusion therapy.
Response 2: Thank you so much for the suggestions. New applications have been added and manuscript updated according to your suggestions. Please check section 5.
Additional Issues:
The introduction is too short and does not adequately set the stage for the review. It should provide more background on albumin, its significance in biomedical research, and a clearer outline of what the manuscript will cover.
Response 3: Thank you so much for the comment. Introduction section has been updated according to your suggestions. Please check section 1.
All figures should be cited and discussed within the text to ensure they are integrated into the manuscript's narrative.
Response 4: Thank you so much for the comment. Figures and tables cited and necessary information provided within the text. Please check tables and figures.
Tables should be provided in an editable format, not as figures, and the authors should clarify whether the tables and figures are original or sourced from other works. Proper citation is essential if they are not original.
Response 5: Thank you so much for the suggestion. Manuscript has been updated according to your suggestion. Please check tables.
Please omit Figure 9, as it does not add value to the manuscript and detracts from the overall focus.
Response 6: Thank you so much for the comment. Figure 9 has been deleted according to your suggestions.
The manuscript would benefit from a thorough review for language clarity and conciseness. There are instances of redundancy and overly technical language that could be simplified for better readability.
Response 7: Thank you so much for the comment. Manuscript has been proofread for language quality.
The conclusion should focus more on summarizing the practical applications and future potential of albumin rather than reiterating the purification techniques.
Response 8; Thank you so much for the comment. Manuscript has been updated according to your suggestion.
Round 2
Reviewer 2 Report
Comments and Suggestions for Authors
Despite the authors' commendable efforts to address previous concerns, I must regrettably recommend rejecting this manuscript due to persistent errors, inconsistencies, and a lack of focus that severely undermine its suitability for publication in its current form.
- Abstract vs. Main Text Discrepancies: The revised abstract describes albumin as a "novel material for the synthesis of nanoparticles for targeted drug delivery." However, this claim is not substantiated with adequate information within the main text. Such a discrepancy misleads readers and detracts from the manuscript's credibility. The content of the abstract should accurately reflect the body of the manuscript, providing a clear and truthful preview of the discussed content.
- Lack of Focus on Relevant Topics: As noted in my initial review, the extensive discussion on albumin purification remains disproportionately extensive and only peripherally relevant to the manuscript’s stated goals. The authors were advised to focus more intensively on the potential biological applications of albumin, e.g., as drug carriers. This would include a detailed exploration of the advantages and disadvantages of albumin compared to other drug delivery systems and a comprehensive review of successful case studies. Unfortunately, these adjustments have not been sufficiently made.
- Quality and Relevance of Visual Aids: The quality of tables and figures presented remains suboptimal, and their integration into the manuscript is poorly executed. For instance, Figure 1, which provides an overview of protein functions, is tangentially related to the manuscript's focus and does not support the central thesis concerning albumin’s role in drug delivery. Moreover, Figure 2 is ambiguous, with legends that fail to provide necessary details such as the Protein Data Bank (PDB) IDs for referenced structures, limiting the figure's usefulness and academic rigor.
- Citations and Scholarly Rigor: Several sections of the manuscript lack appropriate citations, which is particularly noticeable in Section 8, where not a single reference is cited. This omission is a serious scholarly oversight that questions the manuscript's academic foundation and diminishes the trustworthiness of the arguments presented.
The English language needs minor corrections.
Author Response
Reviewer 2 response round 2
1 Abstract vs. Main Text Discrepancies: The revised abstract describes albumin as a "novel material for the synthesis of nanoparticles for targeted drug delivery." However, this claim is not substantiated with adequate information within the main text. Such a discrepancy misleads readers and detracts from the manuscript's credibility. The content of the abstract should accurately reflect the body of the manuscript, providing a clear and truthful preview of the discussed content.
Response: Thank you for your valuable feedback. We acknowledge the discrepancy regarding the mention of nanoparticles in the abstract. The abstract has now been revised to align with the content of the manuscript. Our focus is to provide a balanced overview of the applications, purification methods, and market trends related to albumin. We kindly ask you to review the updated abstract for accuracy. Please check abstract.
2 Lack of Focus on Relevant Topics: As noted in my initial review, the extensive discussion on albumin purification remains disproportionately extensive and only peripherally relevant to the manuscript’s stated goals. The authors were advised to focus more intensively on the potential biological applications of albumin, e.g., as drug carriers. This would include a detailed exploration of the advantages and disadvantages of albumin compared to other drug delivery systems and a comprehensive review of successful case studies. Unfortunately, these adjustments have not been sufficiently made.
Response: Thank you for your valuable input. We have carefully reviewed the purification section and removed unnecessary details to ensure a more focused discussion. Additionally, while the manuscript already covers several novel applications of albumin such as drug delivery, wound healing, antioxidant properties, infusion therapy, COVID-19 therapeutics, tissue engineering, and critical illness, we have now added a concise section specifically addressing the use of albumin as a drug carrier. This section highlights key points to maintain balance and avoid excessive length. Our goal is to present a well-rounded view of albumin’s applications, purification methods, and market trends. Please check section 5.
3 Quality and Relevance of Visual Aids: The quality of tables and figures presented remains suboptimal, and their integration into the manuscript is poorly executed. For instance, Figure 1, which provides an overview of protein functions, is tangentially related to the manuscript's focus and does not support the central thesis concerning albumin’s role in drug delivery. Moreover, Figure 2 is ambiguous, with legends that fail to provide necessary details such as the Protein Data Bank (PDB) IDs for referenced structures, limiting the figure's usefulness and academic rigor.
Response: Thank you for your valuable feedback. We have updated the tables with appropriate references. Additionally, Figures 1 and 2 have been completely removed from the manuscript to ensure better alignment with the central focus.
4 Citations and Scholarly Rigor: Several sections of the manuscript lack appropriate citations, which is particularly noticeable in Section 8, where not a single reference is cited. This omission is a serious scholarly oversight that questions the manuscript's academic foundation and diminishes the trustworthiness of the arguments presented.
Response: Thank you for bringing this to our attention. We have revised the manuscript in line with your suggestions, and Section 8 has been updated with the appropriate citations to ensure scholarly rigor and strengthen the academic foundation of the arguments presented. We kindly request you to review the updated section.
Round 3
Reviewer 2 Report
Comments and Suggestions for Authors
Despite the authors' efforts through two rounds of revisions, I regret to inform you that I cannot recommend the manuscript for publication in its current form. The manuscript continues to suffer from fundamental issues that have not been adequately addressed, impacting its suitability for this journal.
1. Lack of Novelty and Focus: The manuscript persists in focusing extensively on purification methods of albumin and market trends, which diverges significantly from the expected novelty and biological applications critical to our readership. As previously suggested, the authors need to concentrate more on the clinical and biomedical applications of albumin, which are both within the scope of the journal and of greater scientific interest.
2. Abstract and Content Misalignment: The abstract has undergone revisions which unfortunately still do not reflect the true content of the manuscript. Such inconsistencies can mislead readers and detract from the credibility of the work. It is crucial that the abstract accurately summarizes the core focus and findings of the paper.
3. Structural and Citation Issues: The manuscript continues to be marred by disorganized citation sequencing and missing references, notably between lines 284-292. Such lapses not only compromise the academic rigor of the manuscript but also hinder readers’ ability to verify and follow up on the research. Proper and orderly citations are essential for maintaining the integrity of academic work.
4. General Presentation: The manuscript appears poorly prepared, with significant mistakes throughout the text. This does not meet the publication standards expected at this level of scientific communication.
Comments on the Quality of English Language
The quality of English is acceptable.